# Comparative Chloroplast Genomics of *Actinidia deliciosa* Cultivars: Insights into Positive Selection and Population Evolution

**DOI:** 10.3390/ijms26094387

**Published:** 2025-05-05

**Authors:** Xiaojing He, Yang Yang, Xingya Zhang, Weimin Zhao, Qijing Zhang, Caiyun Luo, Yanze Xie, Zhonghu Li, Xiaojuan Wang

**Affiliations:** Key Laboratory of Resource Biology and Biotechnology in Western China, Ministry of Education, College of Life Sciences, Northwest University, Xi’an 710069, China; heexj13@163.com (X.H.); 202021088@stumail.nwu.edu.cn (Y.Y.); 202322556@stumail.nwu.edu.cn (X.Z.); 15287218193@163.com (W.Z.); zhangqijing@stumail.nwu.edu.cn (Q.Z.); 202310286@stumail.nwu.edu.cn (C.L.); 18291091069@163.com (Y.X.)

**Keywords:** *Actinidia deliciosa*, chloroplast genome, positive selection, population evolution

## Abstract

The chloroplast genome, as an important evolutionary marker, can provide a new breakthrough direction for the population evolution of plant species. *Actinidia deliciosa* represents one of the most economically significant and widely cultivated fruit species in the genus *Actinidia.* In this study, we sequenced and analyzed the complete chloroplast genomes of seven cultivars of *Actinidia. deliciosa* to detect the structural variation and population evolutionary characteristics. The total genome size ranged from 156,404 bp (*A. deliciosa* cv. Hayward) to 156,495 bp (*A. deliciosa* cv. Yate). A total of 321 simple sequence repeats (SSRs) and 1335 repetitive sequences were identified. Large-scale repeat sequences may facilitate indels and substitutions, molecular variations in *A. deliciosa* varieties' chloroplast genomes. Additionally, four polymorphic chloroplast DNA loci (*atpF-atpH*, *atpH-atpI*, *atpB*, and *accD*) were detected, which could potentially provide useful molecular genetic markers for further population genetics studies within *A. deliciosa* varieties. Site-specific selection analysis revealed that six genes (*atpA*, *rps3*, *rps7*, *rpl22*, *rbcL,* and *ycf2*) underwent protein sequence evolution. These genes may have played key roles in the adaptation of *A. deliciosa* to various environments. The population evolutionary analysis suggested that *A. deliciosa* cultivars were clustered into an individual evolutionary branch with moderate-to-high support values. These results provided a foundational genomic resource that will be a major contribution to future studies of population genetics, adaptive evolution, and genetic improvement in *Actinidia*.

## 1. Introduction

Actinidiaceae is considered one of the most economically and ecologically important fruit families, which generally includes the genus *Actinidia*, *Clematoclethra*, *Sladenia,* and *Saurauia* species [1]. Molecular phylogenetic studies have revealed that Actinidiaceae and Ericaceae formed sister groups within the asterid clade, having differentiated at 90.5 myr, and the differentiation of *Actinidia* and *Clematoclethra* occurred in the Middle Eocene [2].

*Actinidia deliciosa* is a perennial dioecious deciduous vine, belonging to Actinidiaceae [3]. It is known for its unique flavor, high vitamin C content, dietary fiber, and various minerals, which make it highly nutritious [4]. Due to its superior ecological environment, favorable climatic conditions, and suitable soil pH value, China is the main production area of *A. delicious* [5]. The *A. deliciosa* varieties Xuxiang, Miliang, and Qinmei account for over 80% of the total cultivated area in China [6]. Therefore, *A. deliciosa* is an important germplasm resource for breeding kiwifruit varieties [5].

Due to the species of *Actinidia* having great morphological similarity and the variation often overlapping over species boundaries, the genetic relationship and species limits of *Actinidia* have long been controversial. *Actinidia* was established by British botanist John Lindley in 1836 [7]. According to the morphological characteristics, Dunn divided *Actinidia* into sect. *Vestitae*, sect. *Maculate*, sect. *Ampulliferae,* and sect. *Leiocacarpae* for the first time. Subsequent revisions reorganized these into *Actinidia,* which was divided into sect. *Leiocarpae*, sect. *Maculate*, sect. *Stellatae,* and sect. *Strigosae* [8]. Recent molecular phylogenetic studies have led to the adoption of various methods to explore the species classification of *Actinidia*, yet their results have only partially supported the traditional taxonomic system [9,10].

The plant chloroplast genome, due to its uniparental inheritance, conserved structure, and small size, is particularly suitable for deciphering complicated population evolutionary relationships. Some significant progress has been made in the study of *Actinidia* chloroplast genomes, with their complete sequences now available for multiple species (e.g., *Actinidia eriantha*, *Actinidia hemsleyana*, and *Actinidia chinensis*) [11,12,13]. For instance, studies have demonstrated that the length of the chloroplast genome sequence increased with the chromosome ploidy level in conspecific taxa of *A. chinensis* and *A. deliciosa* [14]. The long repeat sequences, rather than simple sequence repeats(SSRs) in *Actinidia,* were revealed to be the causal agent leading to chloroplast genome size expansion [15]. Genetic evolutionary analysis based on complete chloroplast genomes demonstrated that *A. deliciosa* was clustered closely with *A. chinensis*, *Actinidia melanandra*, and *Actinidia callosa* [16]. However, population genetics studies on different cultivars within a single *Actinidia* species remain entirely unexplored. Several critical questions demand urgent investigation: (1) Do cultivars of the same kiwifruit species exhibit distinctive chloroplast genome evolutionary patterns? (2) Can population evolutionary analyses based on the complete chloroplast genome provide novel insights into the genetic relationships among these cultivated varieties? (3) What is the extent of chloroplast DNA polymorphism and differentiation among cultivars of the same species? Through these investigations, we hope to elucidate the genetic characteristics and evolutionary background of *A. deliciosa* cultivars while enriching molecular datasets for the *Actinidia* genus.

## 2. Results

### 2.1. Chloroplast Genome Features of A. deliciosa Cultivars

The chloroplast genome of *A. deliciosa* is a closed circular double-stranded DNA molecule. The seven newly sequenced *A. deliciosa* varieties' chloroplast genomes ranged from 156,404 bp for *A. deliciosa* cv. Hayward to 156,495 bp for *A. deliciosa* cv. Yate (Figure 1). The structure of these chloroplast genomes was analogous to most chloroplast genomes of plants with a typical quadripartite structure, with two IRs (24,051–24,053 bp) separated by the LSC (87,967–88,057 bp) and SSC (20,331–20,332 bp) regions (Table 1). All seven varieties shared identical GC contents (37.20%), with the LSC (35.50%) and SSC regions (31.10%) showing a lower GC content than the IR regions (42.90%) (Table 1). The high GC percentage in the IR regions was possibly due to the presence of four rRNA genes in these regions. A total of 131 genes (including 18 duplicated genes) were annotated on these chloroplast genomes, comprising 83 protein-coding genes, 40 transfer RNA genes (tRNA), and 8 ribosomal RNA genes (rRNA) (Appendix A). Eighteen genes were duplicated in the IR region, including four protein-coding genes *(rps7*, *ndhB*, *ycf2*, and *ycf15*), ten tRNA genes (*trnA-UGC*, *trnR-ACG*, *trnN-GUU*, *trnH-GUG*, *trnI-CAU*, *trnI-GAU*, *trnL-CAA*, *trnfM-CAU*, *trnM-CAU*, and *trnV-GAC*), and four rRNA genes (*rrn4.5*, *rrn5*, *rrn16*, and *rrn23*). A total of 14 protein-coding genes and 8 tRNA genes contained one or more introns. Seventeen genes contained one or two introns, of which 11 were protein-coding genes (*rps12*, *rps16*, *rp12*, *rp116*, *rpoC1*, *ycf3*, *ndhA*, *ndhB*, *petB*, *petD*, and *atpF*), and 6 were responsible for tRNA genes (*trnA-UGC*, *trnG-UCC*, *trnI-GAU*, *trnL-UAA*, *trnK-UUU*, and *trnV-UAC*) (Appendix A). In general, the structure of chloroplast genomes, such as gene content and gene order, was highly conserved between the seven *A. deliciosa* varieties.

### 2.2. Distribution and Characterization of Repeat Elements

The chloroplast genomes in seven *A. deliciosa* varieties contain numerous palindromic repeats, dispersed repeats (including both forward and reverse repeats), and tandem repeats (Figure 2a and Appendix A). In this study, 1335 repeats were identified, where 528 were dispersed repeats, as the most common of the three types, which accounted for 40% of the total repeats; there were 431 tandem repeats, which accounted for 32%; and there were 346 palindromic repeats, which accounted for 28% (Figure 2b). Notably, the distribution of these three types of repeats in the chloroplast genome of different varieties is highly similar, and they are usually located in the same gene. This large number of repeats may maintain the stability of the chloroplast genome.

Simple repeat sequences (SSRs) exhibit high polymorphism, making them valuable molecular markers that have been widely applied in population genetics and species biodiversity research. Using the MISA-web tool ((MIcroSAtellite identification tool), we analyzed the distribution of various SSR types across seven *A. deliciosa* varieties (Figure 3a). In total, we identified 321 SSRs (including mono-, di-, tri-, tetra-, penta-, hexa-, and polynucleotides), where 237 were found in the LSC region, and 70 and 14 in the SSC and IRb regions, respectively (Appendix A). The mononucleotide repeats were most frequent, comprising 54% of the total, and 13% were tetranucleotide repeats (Figure 3b). The number of trinucleotide repeats was higher than that of dinucleotide repeats, and there were very few pentanucleotide or hexanucleotide repeats in seven *A. deliciosa* varieties. Most of the simple repeat sequences were distributed in the non-coding region, which accounted for 76.00% (intergenic spacer regions = 63% and intron regions = 13%) (Figure 3c). And SSRs located in the coding regions were mainly located in *atpF*, *atpH*, *rpoC2*, *accD*, *atpF, psbI*, *rpoA*, *rpoB*, and *rpoC1* genes. The SSRs identified in the chloroplast genome sequences contained a large number of AT bases, and all mononucleotide and dinucleotide repeats were A/T (Appendix A).

While the total number and distribution of SSRs were largely conserved across the seven *A. deliciosa* varieties (Figure 3a), we identified some SSRs exhibiting length variation or motif alteration across varieties. These variable SSRs were primarily located in the LSC region and the SSC region. For instance, in the intron of the *atpF* within the SSC region, a compound SSR was detected, with its sequence characteristics showing significant differences among different varieties: *A. deliciosa* cv. Nongke-1 possesses a unique compound sequence (T)_10_ctatatctttcta(T)_11_ (34 bp), while all other varieties only exhibited simple (T)_11_ repeats without any intervening sequences. In the *atpF-atpH* spacer within the LSC region, the SSR sequence in most varieties is (T)_10_atatat(TA)_5_ (26 bp), but this SSR locus was completely absent in *A. deliciosa* cv. Hayward. Regarding the trinucleotide SSR in the intron of *atpF* within the SSC region, most varieties (including *A. deliciosa* cv. Hayward) had (TTA)_7_ repeats, but the position of *A. deliciosa* cv. Hayward’s SSR was significantly shifted compared to the other varieties, possibly reflecting genomic structural variation. The compound SSR in the *ycf4*-*cemA* spacer exhibited high discriminative power among varieties: *A. deliciosa* cv. Cuixiang, *A. deliciosa* cv. Yate, *A. deliciosa* cv. Nongda, *A. deliciosa* cv. Mihong: ending with (T)_14_, *A. deliciosa* cv. Xuxiang, and *A. deliciosa* cv. Ximi-59: ending with (T)_12_, with *A. deliciosa* cv. Hayward having a different intermediate sequence compared to other varieties.

### 2.3. Sequence Divergence Patterns of Chloroplast Genomes

We used mVISTA to perform a sequence divergence analysis, with *A. deliciosa* cv. Yate as a reference. Sequence divergence analysis revealed high sequence similarity across the seven *A. deliciosa* varieties' chloroplast genomes (Appendix A), which suggested that the chloroplast genomes were relatively well conserved. In general, the non-coding and single-copy regions exhibited higher levels of divergence than the coding and IR regions, respectively. However, the levels of divergence in hotspot regions were relatively low due to limited intraspecific variation.

The percentage of variation in non-coding regions ranged from 0% to 3.00%, with an average of 0.74%, which was seven times higher than that in the protein-coding regions (0.1% on average) (Appendix A). And two intergenic spacer regions with percentages exceeding 0.80% were *trnG(UCC)*-*trnR(UCU)* and *atpH*-*atpI*. In the non-coding regions, the mean percentages of variations in the LSC, SSC, and IR regions were 0.87%, 0.065%, and 0.00%, respectively, which revealed that the IR region was highly conserved. However, there were only two variation points in the coding region, respectively located in the *atpB* and *accD* gene, and the percentage of variation was 0.1% (Appendix A).

In addition, we compared the expansion and contraction of the LSC/IRs and IRs/SSC borders and their adjacent genes in the chloroplast genomes of seven *A. deliciosa* varieties’ chloroplast genomes (Figure 4). The gene arrangement of all *A. deliciosa* varieties was highly conserved, where *rpl23*, *ndhF*, *ycf1*, *trnH*, and *psbA* were present at the junction of LSC/IRb, IRb/SSC, SSC/IRa, and IRa/LSC. The *ycf1* gene was situated at the junction of IRb/SSC, extending 6635 bp into the IRb region. The *psbA* gene has different positions in these chloroplast genome sequences. In the seven *A. deliciosa* varieties' chloroplast genome sequences in this study, the *psbA* gene was completely localized in the LSC region, and the length to the IRa/LSC boundary was between eight and 14 bp. Notably, the NCBI reference sequence (NC_026690) anomalously places *psbA* entirely within the IRa region, which was a configuration discordant with our data. From the perspective of functional evidence, the plastid gene *psaA* encodes the P700 chlorophyll ɑ apoproteins of PSI and D1 subunits of the core complex of PSII [17]. The plastid-encoded genes for proteins and cofactors associated with the PSI and PSII supercomplexes, such as *psbA*, *psbB*, *psbC*, *psbD*, etc., were scattered across the LSC region [18]. Overall, the structure and gene content of the eight chloroplast genomes were consistent, and no significant expansion or contraction of IR regions was found in the seven *A. deliciosa* varieties.

### 2.4. Positive Selection Genes

We identified six genes with sites under positive selection in the 27 *Actinidia* chloroplast genomes (Appendix A). Interestingly, these genes included one ATP subunit gene (*atpA)*, two small subunits of ribosome genes (*rps3* and *rps7*), one large subunit of the ribosome gene (*rpl22*), one large subunit of the Rubisco gene (*rbcL*), and the *ycf2* gene (Appendix A). In addition, according to the M2 and M8 models, the *ycf2* gene had nine sites under positive selection, while *rbcL* had seven sites, and each of the other four genes had only one active site. Both likelihood ratio tests (M0 vs. M3, M1 vs. M2, and M7 vs. M8) supported the presence of positively selected codon sites (*p* < 0.01).

In order to detect such positive selection sites, we selected the branch-site model for further analysis. The results showed that no positive selection sites were identified when branches other than eight *A. deliciosa* chloroplast genomes were used as foreground branches.

### 2.5. Genetic Evolutionary Analysis

The maximum likelihood (ML) evolutionary tree and Bayesian (BI) evolutionary relationships were constructed based on 27 whole-chloroplast genomes from the Actinidiaceae family using *C. scandens* subsp. *hemsleyi* and *S. tristyla* as outgroups (Figure 5). All nodes in the ML tree had moderate to high bootstrap support values, and these 27 *Actinidia* chloroplast genome sequences were clustered into three major genetic clades. According to the leaf characteristics, *Actinidia* was divided into sect. *Leiocarpae*, sect. *Maculate*, sect. *Stellatae,* and sect. *Strigosae.* Most of the species in the sect. *Leiocarpae* clustered as a paraphyletic group at the bottom of the phylogenetic evolutionary tree. The species of the other three sections were distributed in the other two genetic clades, and the species of these three sections overlap with each other without obvious demarcation. In addition, the results show that the seven *A. deliciosa* varieties in this study and the *A. deliciosa* published on NCBI formed a clade with a high bootstrap support value. This further shows that the seven *A. deliciosa* varieties in this study were stable genetic *A. deliciosa* varieties.

## 3. Discussion

In this study, we sequenced and assembled the complete chloroplast genomes of seven individuals from seven varieties of *A. deliciosa* (Figure 1). The genome size, gene order, and composition in the seven chloroplast genomes analyzed in this study were found to be similar to those previously reported for *A. deliciosa* plastid genomes [19]. The total GC content in the chloroplast genome of different varieties of *A. deliciosa* was 37.2% (Table 1), which was similar to that in most land plants. In addition, the GC content in IR regions was 42.9%, which had the largest difference from that in SSC and LSC regions (31.10% and 35.50%, respectively) (Table 1). This may be due to the existence of four rRNA genes in the IR regions. The highly conserved IR region may be related to the high GC content [20]. Moreover, by comparing the sequence length of each region, it was found that the chloroplast genome length of different varieties of *A. deliciosa* was mainly reflected in the LSC regions.

Repetitive sequences play a vital role in chloroplast genome evolution [21]. The distributions of three repeating types were highly similar in seven *A. deliciosa* chloroplast genomes, and they were usually located in the same regions. Meanwhile, most of the repetitive sequences are distributed in IGS regions. This large number of repeats might contribute to maintaining the stability of chloroplast genomes, and similar results were also obtained in other plant studies [22]. It is very important to detect the correlations between repetitive sequences and single-nucleotide polymorphisms (SNPs), as well as insertions–deletions (InDels) in plant chloroplast genomes. For instance, 88–96% oligonucleotide repeats showed co-occurrence with SNPs at the family and subfamily level, and the extent of correlation ranged from 0.182 to 0.513 between InDels and repetitive sequences in Malvaceae chloroplast genomes [23]. In addition, the narrow correlation between the localization of repetitive sequences and InDels has been reported in complete chloroplast genomes of gymnosperms and angiosperms [24,25,26]. The association between repeats and InDels suggests that repeat sequences can serve as markers for detecting mutational loci.

Building upon the pivotal role of repetitive sequences in shaping chloroplast genome architecture, SSRs represent another class of dynamic elements that contribute to genetic diversity. SSRs are widely distributed in plant chloroplast genomes, which have been widely applied as molecular markers for determining genetic variations across species in evolutionary studies because of their faster evolutionary rates [27,28]. It is noteworthy that a large number of SSRs are distributed in non-coding regions, which may be one of the reasons why the mutation rate in these regions is higher than that in protein-coding regions. Therefore, these SSR units could be used as important molecular markers in populations for addressing genetic diversity among closely related taxa. Some previous studies have demonstrated that polymorphism of SSR might have quantitatively regulated gene transcription [29]. In our study, a total of 322 SSRs were identified across seven *A. deliciosa* varieties, with 44 SSRs being conserved among all cultivars (Appendix A). Overall, the SSRs exhibited low polymorphism levels. The high conservation of these SSRs makes them suitable for cultivar purity testing. We propose the polymorphic SSRs as “negative markers” to exclude non-target species. For instance, *A. deliciosa* cv. Hayward lacked one SSR that was present in all other varieties. Additionally, all types of SSRs were found to be AT-rich (Appendix A), which was consistent with the previous report that poly A and T were the most abundant repeats in most angiosperm chloroplast genomes [30]. AT-rich motifs provide the structural basis for DNA replication slippage [31]. The prevalence of AT-rich sequences in chloroplast genomes likely reflects conserved genetic characteristics inherited from their prokaryotic endosymbiotic ancestor [32]. Interestingly, hexanucleotide SSRs were consistently detected in all seven *A. deliciosa* varieties (Appendix A), while previous reports suggested their exclusive presence in *A. tetramera* and *A. chinensis* chloroplast genomes [13,15,33].

In order to determine the polymorphic loci, we compared the whole cp genome sequences of seven *A. deliciosa* varieties and calculated the percentages of variable characters in coding and non-coding regions (Appendix A). Our results indicated that the chloroplast genomes of seven *A. deliciosa* varieties showed low levels of genetic divergence. Furthermore, the proportion of variable sites was higher in the non-coding regions than the coding regions, which is generally consistent with most previous studies of the plastid genomes of angiosperms [34]. Additionally, four polymorphic loci (*atpF-atpH*, *atpH-atpI*, *atpB,* and *accD*) were identified in the seven *A. deliciosa* varieties’ chloroplast genome, which could be used in phylogenetic analyses or as potential DNA molecular barcodes in future population genetics studies [35].

In the process of genome evolution, the expansion or contraction of the IR regions is an important evolutionary force, which often affects the size variation of different chloroplast genomes [36,37]. In our study, the IR regions showed similar lengths among the seven *A. deliciosa* varieties, ranging from 24,051 bp to 24,053 bp (Figure 4). The results showed that the IR/LSC and IR/SSC boundaries of the chloroplast genome among different varieties might be conserved. Notably, *rpl23* was 162–164 bp to the left of the LSC/IRb boundary, the distance between *rpl23* and the IRb boundary correlated with the length of IRb, and IRb expansion shortened the *rpl23*-IRb boundary distance. In addition, IRa extended into the *ycf1* genes, which was also observed in *Cardiocrinum*, Hamamelidaceae, and numerous other plant species [38,39]. Interestingly, many previous reports have shown that there are also some differences among relatively distantly related species, such as gene overlap length and duplication of the *ycf1* and *rps3* genes, suggesting that the expansion and contraction of IR regions lead to changes in the length and structure of the chloroplast genome [40,41].

In this study, we detected six chloroplast protein-coding genes in the 20 *Actinidia* species that were under positive selection (*atpA*, *rps3*, *rps7*, *rpl22*, *rbcL,* and *ycf2*) (Appendix A). ATP synthase is a ubiquitous enzyme in eukaryotic organelles that is essential for both photosynthesis and respiration [42]. The *atpA* encodes the α subunit of the CF1 complex [43]. Additionally, we detected *rps3* and *rps7* genes, which play an important role in plant chloroplast ribosome synthesis [44]. The *rpl22* gene is a ribosomal protein gene and is one of the common adaptive evolution genes in plant cells, which is mainly involved in the synthesis of ribosomal L22 protein [45]. Furthermore, the *rbcL* gene encodes the large subunit of Rubisco and plays an important role as a modulator of photosynthetic electron transport [46]. Previous studies have indicated that the *rbcL* gene is often under positive selection in land plants [47]. In particular, the *rbcL* gene experienced strong positive selection after the C3–C4 photosynthetic transition [48]. The *ycf2* gene is one of the largest genes in the chloroplast genome, which is a large open reading frame. The role of the *ycf2* gene remains unclear, but the more than 2000 amino acids encoded by *ycf2* in most terrestrial plants are essential for cell survival [49,50]. These positively selected genes may have played important roles in the adaptation of *Actinidia* species to various environments. But the branch-site model did not detect positive selection sites. The result of this may be that, in the long-term evolution process, adaptive evolution occurred in the early stage and has been fixed, or the positive selection sites are covered by a large number of accumulated neutral substitution sites, so the positive selection sites will be difficult to detect [51]. While our analyses employed the widely used PAML with both site-specific models and branch sites, integrating additional approaches and models could provide deeper insights, such as HyPhy [52,53]. Notably, the potential discrepancies between PAML and HyPhy results would not necessarily indicate methodological limitations but might reflect genuine biological complexity.

*Actinidia* presents great obstacles to classification because of the extensive interspecific hybridization and gene introgression [54]. The phylogenetic tree constructed according to the whole-chloroplast genome sequences showed that the four sections divided according to morphology could not be clearly distinguished (Figure 5). The sect. *Leiocarpae* clustered as a paraphyletic group, which was located at the bottom of the phylogenetic tree, and the other three sections of species partially overlapped. Our result is consistent with the classification of *Actinidia* based on the micromorphological characters of foliar trichomes [55]. Li (1952) divided *Actinidia. rufa* into the sect. *Leiocarpae* [8]. However, subsequent studies utilizing chemical composition, genetics, and molecular biology approaches consistently demonstrated that *A. rufa* had not clustered with other species in sect. *Leiocarpae* [9,56]. A study based on isozymes and flavonoids showed that *A. rufa* did not belong to sect. *Leiocarpae*, and it was more reasonable to classify it into sect. *Maculatae* [57]. In conclusion, based on the research results of this paper, we support the hypothesis that *Actinidia* should be divided into sect. *Leiocarpae* and sect. *Maculatae* [5]. This hypothesis needs to be comprehensively analyzed in combination with morphology and phylogeography in the future.

## 4. Conclusions

Through a comparative analysis of the complete chloroplast genomes of seven *A. deliciosa* varieties, this study revealed the population genetics and evolutionary characteristics among cultivated varieties of *A. deliciosa*. The chloroplast genome sequences exhibited extremely high similarity among varieties, with the variations in non-coding regions being significantly higher than those in coding regions. Six positively selected genes were identified, with their selected sites potentially reflecting differential adaptation responses among different cultivated varieties. Trinucleotide, pentanucleotide, and hexanucleotide repeats were rare in *Actinidia* chloroplast genomes. In addition, the precise population relationships of all seven cultivated varieties were determined from *A. deliciosa* for the first time. The seven varieties exhibited certain degrees of genetic differentiation and were primarily clustered into one major evolutionary clade. This study not only enriched the complete chloroplast genome resources of *A. deliciosa* but also provided useful information for further studies of the population evolutionary history of *Actinidia* species. Future research should expand the scope to include additional *Actinidia* species, integrating both chloroplast and nuclear genomics data. Combining genomic information with phenotypic data to explore the origin and population evolution of the *Actinidia* should become a key focus for subsequent evolutionary investigations.

## 5. Materials and Methods

### 5.1. Plant Material and DNA Extraction

Fresh seeds were collected from seven *A. deliciosa* varieties (Cuixiang, XuXiang, Hayward, Nongda Mixiang, Yate, Nongke-1, Ximi-59) in Zhouzhi County, Shaanxi Province, China (Appendix A). Voucher specimens of each sample were deposited in the Key Laboratory of Resource Biology and Biotechnology in Western China (Xi’an, China). The total genomic DNA was extracted using the modified CTAB method [58]. DNA was visualized by 1% agarose gel electrophoresis for quality checks. Subsequently, the complete genomic DNA was subjected to sequencing analysis using the Illumina NovaSeq 6000 sequencing platform, employing paired-end (PE) 150 bp sequencing strategies, which were executed by Novogene Bioinformatics Technology Co., Ltd., Beijing, China. The DNA library was constructed with an estimated mean insert size of 350 bp, and sequencing was carried out to achieve a coverage depth of roughly 50× for each chloroplast genome.

### 5.2. Chloroplast Genomes Assembly and Annotation

The NGSQC Toolkit v2.3.3 program [59] was utilized to filter the original Illumina raw reads, remove low-quality sequences (Phred score < Q20) and adapters, and obtain clean reads for subsequent assembly. We used the reference-guided assembly method to construct the chloroplast genomes with Bowtie v2.4.2 [60]. The chloroplast genome of the closely related species *Actinidia chinensis* (NC_026690) was selected as a reference. Chloroplast genomes were annotated using CPGAVAS2 (http://47.96.249.172:16019/analyzer/home (accessed on 21 April 2023)) to identify genes by using BLAST to search against the custom database [61]. The main various BLAST parameters were as follows: gapped alignment” was set to “yes”, “Genetic Code for Blastx “ was set to “11 plant plasit”, “Percent identity cutoff for protein coding” was set to “60”, “Percent identity cutoff for RNAs” was set to “90”, and “E-value” was set to “1 × 10^−5^”. The preliminary prediction results from DOGMA were aligned with the reference genome (NC_026690) (identity ≥ 95%) to confirm conservation. Based on the DOGMA predictions and the reference genome alignment, we manually adjusted the gene and CDS regions (e.g., correcting start/stop codons and exon–intron boundaries) with Geneious v8.0.2 [62]. Finally, the circular map for seven *Actinidia deliciosa* chloroplast genomes comparison was generated using BLAST Ring Image Generator (BRIG) v 0.95 software [63].

### 5.3. Repeat Elements Analysis

In this study, we examined three repeating types of chloroplast genome sequences, including palindromic, dispersed (forward and reverse repeats), and tandem repeats. The online program REPuter [64] was used to find the dispersed and palindromic repeats based on the following criteria: (1) Hamming distance = 3; (2) sequence identity ≥ 90%; and (3) minimum repeat size = 30 bp. In addition, the tandem repeat sequences were detected using the online program Tandem Repeats Finder (https://tandem.bu.edu/trf/trf.html (accessed on 5 April 2023)) [65], where the alignment parameters match, mismatch, and indel were set to 2, 7, and 7, respectively. The minimum alignment score and maximum period size were 80 and 500, respectively. The simple sequence repeats (SSRs) in chloroplast genomes were identified using the Perl script MISA web (http://pgrc.ipk-gatersleben.de/misa/ (accessed on 14 March 2023)) [66]. The minimum numbers of repeats were 10, 5, 4, 3, 3, and 3 for mono-, di-, tri-, tetra-, penta-, and hexanucleotides, respectively.

### 5.4. Comparative Chloroplast Genome Analysis

To visually assess sequence divergence between chloroplast genome sequences, seven *A. deliciosa* varieties were compared using mVISTA (https://genome.lbl.gov/vista/mvista/instructions.shtml (accessed on 25 March 2023)) [67], with *A. deliciosa* cv. Yate serving as a reference. The percentages of nucleotide variation for coding and non-coding regions were calculated according to the methods of Zhang et al. (2011) [38].

In addition to the seven *A. deliciosa* varieties sequenced in this study, we downloaded a chloroplast genome sequence of *A. deliciosa* (NC_026690) from NCBI. The online program IRscope (https://irscope.shinyapps.io/IRapp/ (accessed on 10 June 2023)) was used to compare expansion and contraction at the IR boundary of 8 *A. deliciosa* chloroplast genomes [68] and draw the SC/IR boundary map among the sequences.

### 5.5. Positive Selection Analysis

In order to detect the sites under selection in the protein-coding genes in *Actinidia* chloroplast genomes, the non-synonymous (dN) and synonymous (dS) nucleotide substitution rates and their ratio (ω = dN/dS) were calculated with Codeml program in PAML v4.7 (seqtype = 1, model = 2, NSsites = 2) [69,70]. Positive selection analysis was conducted based on 27 taxa, including 7 *A. deliciosa* varieties in the current study and 20 other *Actinidia* species downloaded in Genbank format from the National Center for Biotechnology Information database (NCBI, https://www.ncbi.nlm.nih.gov/ (accessed on 11 August 2023)) [71,72] (Appendix A). The protein-coding genes were extracted using Geneious v8.0.2 and aligned using MAFFT v7.0. Maximum likelihood phylogenetic trees were reconstructed based on the complete cp genomes using RAxML v7.2.8 [73].

We employed the site-specific model and branch-site model to analyze the selection pressure based on 74 protein-coding genes shared by 27 *Actinidia* plastomes. The site-specific model allowed the ω ratio to vary among sites, with a fixed ω ratio in all the evolutionary branches. We compared the site-specific models to analyze the existence of selected sites: M0 (one ratio) vs. M3 (discrete); M1 (neutral) vs. M2 (positive selection); and M7 (beta) vs. M8 (beta and ω). The branch-site model A aims to detect positive selection that affects only a few sites on prespecified lineages. The branches being tested for positive selection are called the foreground branches, while all other branches on the tree are the background branches. The log-likelihood ratio test (LRT) [74] was used to estimate the quality of model A. The BEB method is implemented to calculate posterior probabilities for site classes under model A if the LRT suggests presence of codons under positive selection on the foreground branch.

### 5.6. Phylogenetic Analysis

Phylogenetic relationships were reconstructed based on 29 taxa, including 7 *A. deliciosa* varieties in the current study, 20 other *Actinidia* species, and 2 Actinidiaceae species *(Clematoclethra scandens* subsp. *hemsleyi* and *Saurauia tristyla*) that were used as outgroups (Appendix A). The phylogenetic trees were reconstructed based on the complete chloroplast genomes. First, all the chloroplast genome sequences were aligned with MAFFT v7.0 [75].

jModelTest v2.1.10 [76] was used to determine the best-fitting model. Finally, maximum likelihood analysis was conducted using the program RAxML v7.2.8 with the GTR+G model for 1000 replications [73]. Bayesian inference was conducted using the program MrBayes v3.2.2 [77]. Markov chain Monte Carlo simulations were independently run twice for 1 million generations, and sampling trees every 1000 generations. Convergence was determined by examining the average standard deviation of split frequencies. During the operation of the algorithm, the first 25% of trees were discarded as burn-in.

## Figures and Tables

**Figure 1 ijms-26-04387-f001:**
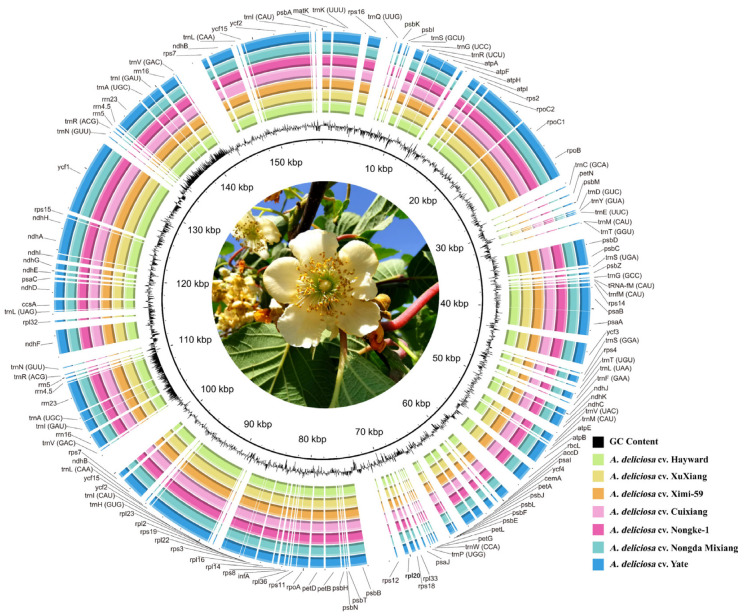
Comparison of chloroplast genomes of seven *A. deliciosa* varieties using BRIG. The sequence of *Actinidia chinensis* (NC_026690) is selected as reference, and the innermost ring shows GC content.

**Figure 2 ijms-26-04387-f002:**
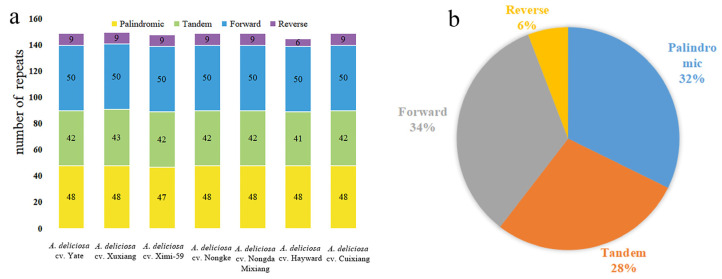
Analysis of repeated sequences in seven *A. deliciosa* varieties' chloroplast genomes. (**a**) Bar graph indicating the numbers of four repeat types (palindromic repeats, tandem repeats, forward repeats, and reverse repeats) in each individual. (**b**) Pie chart revealing the proportion of different repeated sequence types in seven *A. deliciosa* chloroplast genomes.

**Figure 3 ijms-26-04387-f003:**
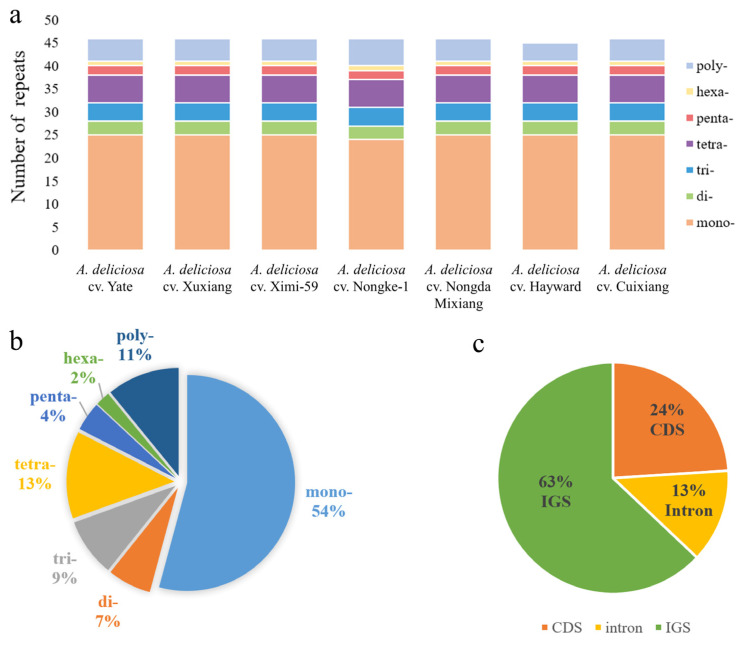
Analysis of simple sequence repeats (SSRs) in seven *A. deliciosa* varieties’ chloroplast genomes. (**a**) Bar graph indicating the number of different SSR types detected in each individual, including mononucleotide repeats (mono-), dinucleotide repeats (di-), trinucleotide repeats (tri), tetranucleotide reapts (tetra-), pentanucleotide repeats (penta), hexanucleotide repeats (hexa-), and polynucleotide repeats (poly-). (**b**) Pie chart revealing the proportion of different SSR types in seven *A. deliciosa* chloroplast genomes. (**c**) Pie chart revealing the frequency of SSRs in the intergenic spacer regions (IGS), protein-coding genes (CDS), and introns.

**Figure 4 ijms-26-04387-f004:**
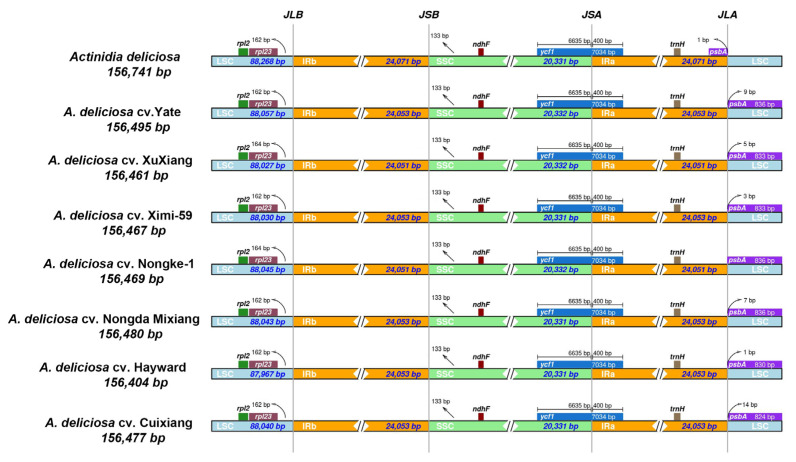
Comparison of the borders of the LSC, SSC, and IR regions among *A. deliciosa* chloroplast genomes. For each variety, genes transcribed in positive strand are depicted on the top of their corresponding track from right to left, while the genes on the negative strand are depicted below from left to right. The numbers at arrows refer to the distance of the start or end position of a given gene from the corresponding junction site. The T bars above or below the genes indicate the extent of their parts, with their corresponding values in the base pairs. The plotted genes and distances in the vicinity of the junction sites are the scaled projection of the genome. JLB (IRb/LSC), JSB (IRb/SSC), JSA (SSC/IRa), and JLA (IRa/LSC) denote the junction sites between each corresponding two regions of the genome.

**Figure 5 ijms-26-04387-f005:**
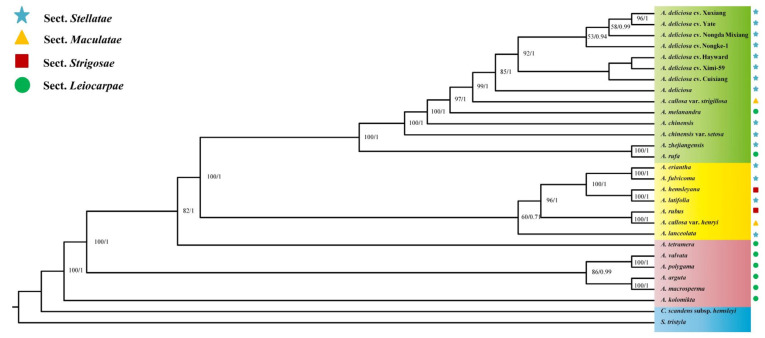
Phylogenetic tree of the *Actinidia* species constructed via maximum likelihood (ML) and Bayesian inference (BI) by using whole-chloroplast genomes. The numbers to the left of the slashes on the branches represent the bootstrap values obtained from ML analysis with the GTR+G model for 1000 replications, while those to the right represent the posterior probabilities derived from BI with MCMC algorithm, with 1,000,000 generations, sampling every 1k, and 25% burn-in.

**Table 1 ijms-26-04387-t001:** Summary of chloroplast genome characteristics of *A. deliciosa* varieties.

Genome Features	*A. deliciosa* cv. Cuixiang	*A. deliciosa* cv. XuXiang	*A. deliciosa* cv. Hayward	*A. deliciosa* cv. Nongda Mixiang	*A. deliciosa* cv. Yate	*A. deliciosa* cv. Nongke-1	*A. deliciosa* cv. Ximi-59
Size (bp)	156,477	156,461	156,404	156,480	156,495	156,479	156,467
LSC length (bp)	88,040	88,027	87,967	88,043	88,057	88,045	88,030
SSC length (bp)	20,331	20,332	20,331	20,331	20,332	20,332	20,331
IR length (bp)	48,106	48,102	48,106	48,106	48,106	48,102	48,106
Coding regions (bp)	76,941	76,939	76,395	76,941	76,941	76,939	79,941
Non-coding regions (bp)	79,536	79,522	80,009	79,539	79,554	79,540	76,526
Number of genes	131	131	131	131	131	131	131
Protein-coding genes	83	83	83	83	83	83	83
tRNA genes	40	40	40	40	40	40	40
rRNA genes	8	8	8	8	8	8	8
GC content (%)	37.20	37.20	37.20	37.20	37.20	37.20	37.20
GC content of LSC (%)	35.50	35.50	35.50	35.50	35.50	35.50	35.50
GC content of SSC (%)	31.10	31.10	31.10	31.10	31.10	31.10	31.10
GC content of IR (%)	42.90	42.90	42.90	42.90	42.90	42.90	42.90

## Data Availability

Chloroplast genomes of *A. deliciosa* are being prepared for submission to NCBI.

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
