# Peer review of "Comparative Chloroplast Genomics of Actinidia deliciosa Cultivars: Insights into Positive Selection and Population Evolution"

_ijms, 2025, doi:10.3390/ijms26094387_

Round 1

Reviewer 1 Report

Comments and Suggestions for Authors

Dear Authors,

I have carefully reviewed your manuscript on comparative chloroplast genomics of Actinidia deliciosa. The paper is generally well-written and presents interesting findings. However, I have some minor concerns that should be addressed before publication:

Methods section:
Please provide more details about the sequencing platform and parameters used
Clarify the criteria used for genome annotation validation
Include version numbers for all software tools used in the analysis
Results and Discussion:
Some figures could benefit from improved resolution and labeling
Consider adding statistical support values to the phylogenetic trees
Provide more detailed discussion of the comparative IR boundary variations observed
Explain the biological significance of the identified SSRs in greater detail
Technical corrections:
Check for consistency in species name formatting throughout
Several references need complete journal names
Some figure legends need more detailed descriptions
Minor grammatical errors should be corrected (specific locations marked in manuscript)
Additional suggestions:
Consider adding a brief section on potential applications of the findings
The conclusions could be strengthened with more specific future research directions
A graphical abstract would help readers quickly grasp the main findings
Overall, this is a valuable contribution that will be suitable for publication after addressing these minor revisions. The comparative genomic analysis is thorough and the findings advance our understanding of chloroplast genome evolution in Actinidia.

Author Response

We sincerely appreciate your time and effort in reviewing our manuscript and for providing us with constructive comments and suggestions. Your insightful feedback has significantly improved the quality of our work. We have thoroughly analyzed and addressed each of your suggestions, and provided detailed written responses to all points. Please see the attachment.

Reviewer 2 Report

Comments and Suggestions for Authors

I have provided very detailed comments in the file attached. The main points that I want to highlight are as follows: 
1. The write-up needs improvement at some points. 
2. The presentation of the article should be improved through the article. 
3. The introduction should be improved, and the main research questions should be mentioned after citing relevant literature on the genus. 
4. There are some discussions, but the results are not present. 
5. Some results are shown to have very high value, but there is nothing new, for example, SSRs and repeats. 
6. The discussion should be rewritten, and the results and discussion should be very clear. 
7. The tRNA analysis section should be clarified or removed from the article. We don't understand why these analyses are done or how they broaden our knowledge. 

Comments on the Quality of English Language

The write-up needs improvement for grammar and clarity. 

Author Response

(The authors gave the same response as above.)

Reviewer 3 Report

Comments and Suggestions for Authors

Comments

The manuscript titled “Comparative chloroplast genomics of Actinidia deliciosa: insights into positive selection, phylogeny and variation of chloroplast tRNAs” is informative. The authors have sequenced chloroplast genomes of several accessions of Actinidia deliciosa and carried out basic assessments. Below are some comments for the authors to address.

Major comments

  1. Each figure is expected to be understood independently (at least for the experts) with little to no help from the main text. However, some of the figures contain too little information in their legend. This particularly applies to Figure 5. The authors are kindly suggested to include the models they used, bootstrap value, and any other necessary information in its legend.
  2. The manuscript would benefit significantly if it were to be edited thoroughly for its written English. The authors are kindly suggested to do so.

Minor comments

  1. Page 1, line 18: “relatyionships” => “relationships”

Comments on the Quality of English Language

The manuscript would benefit significantly if it were to be edited thoroughly for its written English. The authors are kindly suggested to do so.

Author Response

(The authors gave the same response as above.)

Round 2

Reviewer 2 Report

Comments and Suggestions for Authors

I think the research question can be improved further. In addition, I think emphasis on phylogeny is not very accurate. Authors need to describe previous work with relevant literature and don't ignore the recent articles relevant to the family. They then develop a research question and focus on it. In current the article is very descriptive. 

Author Response

We sincerely appreciate your time and effort in reviewing our manuscript and for providing us with constructive comments and suggestions. We have thoroughly analyzed and addressed your suggestions, and provided detailed written responses to all points. Please see the attachment.

Reviewer 3 Report

Comments and Suggestions for Authors

Comments

The authors have addressed the all of the comments offered for the initial version of the manuscript and have now made significant changes to it. Below are some additional comments for minor changes-

  1. Page 5; line 131: “roportion” => “proportion” (?)
  2. Page 6; line 154 and similar others: The cases like “(T)10ctatatctttcta(T)11” can be better written with the number values as the subscripts.

Author Response

We sincerely appreciate your time and effort in reviewing our manuscript and for providing us with constructive comments and suggestions. We have  provided detailed written responses to all points. Please see the attachment.

Round 3

Reviewer 2 Report

Comments and Suggestions for Authors

I think the manuscript is close to acceptance. I have some comments. 

The authors should remove the following lines from the abstract:

The population evolution and genetic relationships of Actinidia species have long been controversial due to the high morphological variability and the frequent gene flow and/or genetic introgression among different species and/or cultivars.

 If they want to include these lines, they should change the whole statement instead of just adding some additions for population genetics. The statement confuses population genetics and phylogenetics. 

I appreciate the hard work of authors. They can look further at other parts.

The figure 4 visualization is not good. The ndhF is not represented well. The number crossing the other border. They can do manual corrections, as IRscope always produces minor visual issues. At the JSB junction, none of the genes were shown in the IRb, which is not a correct representation. Similarly, the visualization of trnH is not good, and we don't know how far away the gene is present. They can look some other good article and see to the figure representations of different junctions. 

From lines 256-263: The Authors discussed the role of repeats. I like their discussion, but providing much focus on inversion is not good, as the authors did not discover inversion events at the genus level. The authors discuss the reason for the high mutational rate due to repeats, but do not include the recently published articles that show the correlations of repeats with SNPs and insertion-deletions (Indels) based on the comparative analysis of complete chloroplast genomes. Broaden the discussion in the light of those articles which reported the role of repeats in the generation of SNPs and Indels based on comparison of complete genomes.

I also don't understand why they mention specific genes in this discussion. Did they find any specific repeat abundance for the gene or intergenic spacer regions of the gene?

line 299-307: The authors mention the divergence hotspot. I think the hotspot region definition is slightly different in the literature and is used for a specific level of nucleotide diversity. As the nucleotide diversity of the identified regions is not very high. So, I will suggest the identification of polymorphic loci. The authors should consider these points for all parts of the manuscript and make modifications. 

Identifying positively selected genes based on only one method is not enough. The authors should check other models. They can look at the article below for reference. 

Complete Chloroplast Genomes of Anthurium huixtlense and Pothos scandens (Pothoideae, Araceae): Unique Inverted Repeat Expansion and Contraction Affect Rate of Evolution

Finally, the conclusion should also mention the genetic aspect of the population. In addition, they can make a separate section for the conclusion. 

A quality revision of the conclusion will be good, as it looks like a repetition of the abstract. The abstract should also be changed a little. 

Author Response

(The authors gave the same response as above.)
